# Modic Changes as Biomarkers for Treatment of Chronic Low Back Pain

**DOI:** 10.3390/biomedicines13071697

**Published:** 2025-07-11

**Authors:** Jeffrey Zhang, Emily Bellow, Jennifer Bae, Derek Johnson, Sandi Bajrami, Andrew Torpey, William Caldwell

**Affiliations:** 1Renaissance School of Medicine, Stony Brook University, Stony Brook, NY 11794, USA; jeffrey.zhang@stonybrookmedicine.edu (J.Z.); emily.bellow@stonybrookmedicine.edu (E.B.); jennifer.bae@stonybrookmedicine.edu (J.B.); derek.johnson@stonybrookmedicine.edu (D.J.); sandi.bajrami@stonybrookmedicine.edu (S.B.); 2Department of Anesthesiology, Stony Brook University Hospital, Stony Brook, NY 11794, USA; andrew.torpey@stonybrookmedicine.edu

**Keywords:** modic change, basi-vertebral nerve, patient-reported outcomes, chronic low back pain

## Abstract

**Background:** Chronic low back pain (CLBP) is the leading cause of disability both within the United States and globally. However, reliable diagnosis and treatment remains limited due to a lack of objective and image-based biomarkers. Modic changes (MCs) are visible vertebral endplate and bone marrow changes in signal intensity seen on MRI. MCs have emerged as promising correlates with degenerative disc disease and CLBP. **Methods:** This is a non-systematic literature review. **Results:** This review synthesizes current evidence on the classification, pathophysiology, and imaging of MCs, with a particular focus on their associations with patient-reported outcomes, including pain (Visual Analog Scale), functional status (Oswestry disability index and Roland-Morris Disability Questionnaire), and health-related quality of life (Short Form-36 and EuroQol 5-Dimension 5 Level). MC type 1 and 2 show significant correlations with symptom severity and predict positive response to basi-vertebral nerve (BVN) ablation, a minimally invasive intervention inhibiting the nerves’ ability to transmit pain signals. **Conclusions:** Across multiple trials, BVN ablation has shown significant sustained improvements in patient-reported outcomes among patients with MC, reinforcing their role as both a diagnostic and therapeutic biomarker.

## 1. Introduction

Chronic low back pain (CLBP), defined as pain in the lumbar region for greater than 3 months, affects an estimated half a billion people worldwide and represents the leading global cause of years lived with disability [1]. Despite the prevalence of CLBP, objective biomarkers that correlate clinically with CLBP syndromes have remained elusive. As a result, patients with CLBP can face diagnostic uncertainty and treatment delays, contributing to poor outcomes. Modic changes (MCs), first described by Michael Modic in 1988, are vertebral endplate and bone marrow changes visible on MRI that correlate with degenerative disc disease (DDD) and CLBP [2]. MCs are classified into three distinct types based on their MRI appearance and underlying pathophysiology: type 1 (edematous), type 2 (fatty), and type 3 (sclerotic) [3]. The pathogenesis of MCs is likely related to mechanical damage to the vertebral endplates, which triggers an inflammatory response that leads to the characteristic bone marrow changes on imaging [4]. Studies have demonstrated a correlation between MC and activity limitation in CLBP patients, demonstrating the potential utility of MC as biomarkers in guiding the diagnosis and treatment of CLBP [5]. Basi-vertebral nerves (BVNs) provide nociceptive input to vertebral endplates and demonstrate extensive signaling in areas of discogenic disease [6,7]. As a result, BVNs have emerged as a therapeutic target in the treatment of CLBP, and MCs have emerged as key biomarkers that serve as criteria for BVN ablation, a minimally invasive treatment for CLBP [8]. In this review of the literature, we examine the role of MCs as biomarkers in the diagnosis and treatment of CLBP, with a focus on the relevant epidemiology, imaging acquisition methods, underlying pathophysiology, patient-reported measures, and the role of MC in interventional treatment for CLBP.

## 2. Back Pain Epidemiology

Low back pain (LBP) is the leading cause of disability in the United States, accounting for 4.3 million years lived with disability annually-nearly twice the burden of any other health condition [9]. CLBP affects approximately 13% of adults, with one-third of those individuals experiencing moderate to severe pain [10]. CLBP is the primary contributor to long-term disability, morbidity, and high healthcare and societal costs. In contrast, acute LBP, which typically resolves within a few weeks, often receives less attention due to its generally favorable prognosis [11]. LBP is characterized by pain, muscle tension, or stiffness in the area between the lower rib margin and gluteal folds, which may or may not be accompanied by leg pain [12].

While most cases of acute LBP resolve without intervention, the majority of individuals experience recurrent episodes within a year. Specific LBP refers to symptoms that can be traced to an identifiable underlying condition, such as vertebral compression fractures, osteoporosis, infections, or inflammatory disorders such as rheumatoid arthritis [12]. In contrast, non-specific LBP describes cases where no distinct pathological cause can be determined. Approximately 90% of individuals with LBP fall into this category, with the diagnosis being made after ruling out identifiable medical conditions [12]. In Croft’s prospective study, only 25% of patients who sought care for LBP had completely recovered within 12 months [13]. Although 90% of patients stopped seeking medical attention within three months, the majority continued to experience significant pain and functional limitations [13]. This suggests that LBP can no longer be characterized as a sequence of acute problems but rather as a chronic disease with acute episodes, exacerbations, and recurrences. In Hides’ study, the cumulative risk of at least one recurrence within 12 months varied from 66% to 84% [14].

Although lumbar disc degeneration is almost universally present in patients with symptomatic LBP, its direct contribution to pain remains uncertain, as similar degenerative changes are frequently seen in asymptomatic individuals [15]. The slow and irreversible nature of DDD, combined with the episodic and recurrent presentation of LBP, complicates the understanding of its etiological significance. Degenerative processes primarily target the nucleus pulposus of the intervertebral disc, leading to decreased proteoglycan levels, reduced water content, and subsequent loss of disc height. These changes compromise the disc’s ability to maintain its mechanical function, disrupting normal load transmission and making the spine more vulnerable to injury [16]. Additionally, degenerative intervertebral disc disease can manifest through various symptoms, such as axial back pain, spinal stenosis, myelopathy, or radiculopathy [16].

MCs, identifiable as vertebral endplate and bone marrow alterations on MRI, are increasingly recognized for their association with intervertebral disc degeneration. In their 2016 study, Maatta et al. found that MCs are significantly associated with various components of disc degeneration, including loss of disc height and changes in disc signal intensity [17]. The study also highlighted a correlation between MC and increased body mass index, suggesting that mechanical loading and metabolic factors may contribute to the development of these vertebral endplate alterations. These findings highlight the critical need to consider MC when evaluating and managing patients with DDD and LBP.

Hu et al. propose that MC may play a key role in the initiation and progression of lumbar intervertebral disc degeneration [18]. They suggest that MCs contribute to disc herniations by compromising the integrity of the disc, hindering its natural repair mechanisms, and increasing the risk of extrusion [18]. MCs may also affect the mechanical load distribution on the intervertebral disc and disrupt the vascular structure of the vertebral endplate, impairing nutrient exchange between the vertebrae and the disc [18]. This disruption compromises the disc’s metabolic processes and accelerates degeneration.

## 3. Description of Modic Change, MRI Acquisition Technique, Staging

MCs are vertebral end plate changes seen on MRI, associated with chronic inflammation. These changes were classified in 1988 by Modic et al. into types 1 and 2, along with corresponding pathological change; Modic et al. later described type 3 changes as well [2,19]. “Mixed” types have also been described, representing transition states between Modic types [2,20,21].

Type 1 MCs are characterized by decreased signal intensity on T1-weighted imaging and increased signal intensity on T2-weighted imaging [2,5,21,22,23,24,25,26,27,28]. This corresponds with edematous [20,23,25,28]/fibrovascular [2,21,22,23,26,27] changes in the vertebral end plate, indicating inflammation [5,21,23,25,28].

Type 2 MCs are defined by high signal intensity in T1-weighted imaging but iso- or high-intensity signaling on T2 imaging, representing replacement of hematopoietic marrow with fatty tissue [2,5,20,21,22,23,24,25,26,27,28].

Type 3 MCs are identified as showing decreased intensity on both T1- and T2-weighted imaging and represent sclerotic changes [5,20,21,22,23,24,25,26,27,28].

Mixed-type MCs occur when features of more than one of types 1–3 are present, with the predominant features listed first (e.g., type 1/2 represents primarily type 1 but with some type 2 characteristics) [20].

These are the general radiological features of MC; however, variation in imaging protocol exists and can impact identification of these changes. For example, choice of magnetic field strength can affect the MC seen: more MC overall and more Modic type 2 changes but fewer type 1 changes were diagnosed in a 1.5 T scanner compared with a 0.3 T scanner [5,21]. Dagestad et al. showed that STIR and DixonT2w techniques are interchangeable for evaluation of MC edema [29] and, likewise, Yang et al. showed that using T2-weighted Dixon water only/fat only imaging is equivalent to T1/T2 as far as identification of MC is concerned [30].

T1 MRI techniques utilized for assessment of MC include fast spin-echo [21,26,28,30,31], T1 FLAIR [22], and STIR [29,31]. On the other hand, T2 MRI techniques include fast spin-echo [20,21,22,26,28,30,31], Dixon [29,30], STIR [26], and FLEX [30].

In validation of the distinction between edematous type 1 changes and fatty type 2 changes, Fields et al. used iterative decomposition of water and fat with echo asymmetry and least-squares estimation (IDEAL) to measure the bone marrow fat fraction (BMF) of subjects with Modic type 1, type 2, or no changes as evaluated by traditional T1/T2 imaging [31]. They found a statistically significant reduction in BMF in endplates with type 1 changes compared to controls (absolute difference −22.3%; 95% CI −24.8, −19.8; *p* < 0.001), while type 2 endplates had significantly elevated BMF relative to controls (absolute difference 21.0%, 95% CI 19.1, 22.9; *p* < 0.001) [31].

Other MRI biomarkers for low back pain have been studied. Disc degeneration has been shown to have a high negative predictive value (−98%) but a low positive predictive value for low back pain. Small disc herniations are common in asymptomatic individuals, whereas large herniations typically cause radicular pain rather than low back pain. Facet joint arthritis has a low positive predictive value for low back pain, with as many as 12–61% of patients with these changes on MRI being asymptomatic. T2 hyperintensities in the posterior annulus fibrosis, or “high intensity zones,” have had varied sensitivity (27–81%), specificity (79–97%) and positive predictive value (53–95%) among different studies. In contrast to these examples, Modic changes have consistently shown high specificity and positive predictive value for low back pain [32].

## 4. Underlying Pathology of Modic Change

MC represents an array of vertebral end-plate and subchondral responses to chronic insult. MCs were first described radiographically, but their biology is now understood as an organized cascade that begins with mechanical injury and associated inflammation-driven bone marrow remodeling resulting in sclerosis.

Repetitive axial loading, shear stress, and micro-fractures of the thin cartilaginous vertebral endplate disrupts its function as a barrier. This dysfunction permits nucleus pulposus derived proteoglycans, damage associated molecular patterns, and degraded collagen fragments to diffuse into highly vascularized subchondral marrow [33]. In vitro studies have shown that these molecules activate Toll-like receptors on osteoblasts and marrow macrophages, which primes a pro-inflammatory state that is characteristic of type 1 MC.

The initial injury triggers release of TNF-α, IL-1β, IL-6, and prostaglandin E_2_ from macrophages and local marrow stromal cells [34]. Elevated TNF-α upregulates vascular endothelial growth factor, resulting in marrow edema that is visible as a T2 hyperintensity. Histologic studies have shown that new capillary buds and immunoreactive nerve fibers penetrate the damaged end-plate. These nerve fibers express neurotrophic factors such as Trk-A, Trk-B, and Trk-C. Their proximity to nociceptive neuropeptides contribute to heightened pain sensitivity and elevated Visual Analogue Scale (VAS) scores seen in these patients [35].

As the pathology progresses, hematopoietic cells undergo adipogenic conversion under the influence of peroxisome proliferator activated receptor γ. BMF studies confirm this stepwise progression from type 1 to type 2 lesions, which is accompanied by downregulation of pro-inflammatory cytokines and upregulation of adipokines such as leptin [33]. This altered lipid cascade may impair osteoblastic differentiation, which explains trabecular thinning and relative mechanical weakness associated with type 2 MC.

Resorptive promoting cytokines dominate early MC, stimulating osteoclast genesis and creating marrow cavities. As the initial inflammation subsides, a repair phase ensues: osteoblasts deposit woven bone, transforming cavities into sclerotic patches characteristic of type 3 MC [36]. These patches are imperfect substitutes for original bone. They exhibit reduced perfusion and diminished nerve density. This is clinically correlated with lower pain intensity than in earlier stages of MC.

Chronic inflammatory states in patients with diabetes, metabolic syndrome, and those who smoke can create a systemic low grade inflammatory background. These states augment local cytokine responses and accelerate fatty conversion. Additionally, genetic polymorphisms in IL-1α and IL-1RN have been found to be associated with a two-fold increase in Type 1 MC prevalence. Other polymorphisms such as in MMP-3 and VDR predispose to Type 2 MC [37]. These variants are thought to be involved in extracellular matrix turnover and lipid metabolism. The combined findings suggest a two-hit hypothesis for the development of MC associated with CLBP.

Disc degeneration adjacent to MC produces three- to five-fold higher levels of IL-6, IL-8, and TNF-α compared to discs of equivalent Pfirrmann grade without MC. These cytokines fuel a cycle in which disc catabolism and end-plate inflammation reinforce each other. Over time, this cycle manifests as vertebrogenic pain, defined as flexion evoked, midline lumbar pain that is correlated with Type 1 and Type 2 lesions. This type of CLBP is responsive to BVN ablation. As the process progresses, Type 3 sclerotic changes dominate, which clinically corresponds to reduced pain, but persistent stiffness.

Over time, microfractures at the vertebral endplate elicit nucleus-pulposus proteoglycans and collagen fragments to infiltrate the subchondral marrow. The ensuing Toll-like receptor activation initiates an inflammatory cascade including TNF-α, IL-1, and IL-6, creating the edematous and metabolically active milieu characteristic of Type 1 MC. This is the underlying mechanism that results in chronic low back pain [38]. What follows is a typical reaction to chronic inflammation, including neovascularization and nociceptive nerve in growth across the breached vertebral end plate [25]. This neurovascular development leads to hyperalgesia. As inflammation progresses, lipid rich Type 2 zones appear, which contain weakened trabeculae and leptin-sensitized nociception [33]. Therapeutically, these mechanisms have yielded targeted interventions such as basi-vertebral nerve ablation, which shows durable reductions in pain and disability. Pharmacotherapy often includes NSAIDs, but patients report mixed outcomes. Directed pharmacotherapy currently remains limited. There is potential for targeting the inflammatory and leptin-centric mechanisms behind MC 1 and 2 with on-market drugs that are indicated for other disease processes. For example, glucagon-like peptide-1 receptor agonists (GLP-1 RAs) downregulate TNF-α, IL-1β, and IL-6, and lower high-sensitivity CRP, thus dampening cytokine burst that occurs in Type 1 changes [39]. GLP-1 RAs have also been found to divert mesenchymal progenitors away from PPAR-γ mediated adipogenesis toward an osteogenic fate, which can possibly counteract the fatty conversion and trabecular thinning found in Type 2 changes [40]. Finally, GLP-1 RAs inhibit central nociceptive transmission and oxidative stress, presenting a direct pathway by which they can alleviate the hyperalgesia found in advanced stage Modic changes [41].

## 5. Modic Changes and Patient-Reported Measures

Research has explored the relationship between MC and patient-reported outcomes (PRO), particularly in individuals with CLBP. Patient-reported outcome measures capture patients’ subjective symptoms, functional status and health-related quality of life, providing critical insight to patient experiences [42,43,44,45]. This is particularly valuable in spinal MC, where MRI findings may not directly reflect symptom severity. This section synthesizes findings from the literature on how MCs correlate with PRO using standardized measures, such as the VAS, Oswestry Disability Index (ODI), Roland Morris Disability Questionnaire (RMQ), Short Form 36 (SF-36), and EuroQol 5-Dimension 5 Level (EQ-5D-5L) [42,43,44,46].

### 5.1. Pain Intensity

VAS is a widely used measure for the assessment of pain intensity and has been validated as a reliable and appropriate tool for clinical use [47,48]. Several studies have demonstrated that specific MC types are associated with higher pain scores. Teraguchi et al. studied 814 patients with LBP undergoing lumbar MRI and found that patients with Modic type 1 or mixed Modic type 1/2 reported significantly higher VAS scores than those without MC [49]. Notably, patients with mixed Modic type 1/2 changes had significantly greater VAS scores than in those with only Modic type 1 change, suggesting a compounded clinical impact [49]. Other studies are more variable. Baker et al. retrospectively analyzed 861 patients undergoing anterior cervical discectomy and fusion, including 356 with MC [50]. Patients with and without MC had similar VAS scores, suggesting that MCs do not independently worsen any PRO following anterior cervical discectomy and fusion [50].

Interventional studies offer a different perspective. Kim et al. investigated 14 patients with CLBP and Modic type 1 or 2 changes unresponsive to conservative care who then underwent BVN ablation in a transformational epiduroscopic approach [51]. Significant reductions in VAS scores were observed 3 months post-operatively compared to their baseline. These reductions were sustained throughout up to their final follow-up 12–20 months later, all with *p*-values < 0.0001 [51]. This suggests that epiduroscopic BVN ablation can achieve significant VAS pain reduction in patients with MC type 1 or 2 changes with CLBP.

### 5.2. Patient Function

ODI and RMQ have emerged as the most common and recommended measures to assess function in patients with spinal disorders [52,53]. RMQ is better suited in primary care settings to obtain follow-up at low costs because it can easily be administered over the phone [54]. Conversely, ODI may be more useful in specialty care or when a disability is likely to remain high throughout a trial [52]. In either instance, both the ODI or RMQ are well-validated measures and highly acceptable for use in practice [54].

Albert et al. measured RMQ in 29 patients with Modic type 1 changes who had previously participated in a randomized controlled trial comparing two types of lumbar herniated disc treatment [44]. Prior to Amoxicillin-clavulanate (500 mg/125 mg) treatment, patients in this cohort were observed to have worsening RMQ scores over time [44]. Following administration of antibiotic treatment, approximately two-thirds of patients achieved a clinically meaningful (>30%) improvement in RMQ scores from baseline [44,55]. This trial suggests that bacterial infection may play a role in worsening patient function in those with LBP and MC.

In a longitudinal study, Mitra et al. performed a prospective MRI study of 44 patients consisting of 48 lumbar levels with Modic type 1 endplate changes [56]. Patients were administered a lumbar MRI and assessed for pain and disability using the VAS and ODI scales, respectively. At follow-up, 18 lesions fully converted to type 2 changes, 7 lesions partially converted to type 2, 19 increased in type 1 severity, and 4 remained unchanged [56]. Patients whose lesions increased in severity had on average higher mean VAS and ODI (5.7 and 42.3, respectively), while lower scores were seen in those whose lesions converted fully to type 2 MC (3.8 and 27, respectively). However, these changes did not reach statistically significant levels (*p* = 0.16 for VAS and *p* = 0.09 for ODI) [56].

### 5.3. Health Related Quality of Life

The Short-Form Health Survey is adequate in measuring the physical and mental components of quality of life separately, but concerns have been raised regarding its suitability as a composite index [57]. However, it remains the most widely used instrument to access both physical and mental dimensions of a patient’s health-related quality of life [58]. The 5-level EuroQol 5-Dimension is a more recent iteration of the EQ-5D tool, which assesses health based on mobility, self-care, usual activities, pain/discomfort, and anxiety/depression [59]. The old EQ-5D has a history of use in a wide range of populations, while the newer EQ-5D-5L is still reliable but remains relatively untested [60]. Both the SF-36 and EQ-5D-5L are never used alone; they are always used in conjunction with the other measures mentioned above.

### 5.4. Multiple Patient-Reported Measures

Some studies utilize multiple measures to comprehensively assess treatment impact. For example, in a cohort study by Chen et al., patients with and without MC received nonsurgical treatment for LBP [61]. Three months after nonsurgical treatment, all patients showed a statistically significant improvement in both VAS and ODI scores in all groups (*p* = 0.023, 0.014). Over the next three months, patients with Modic type 1 changes continued to show significant improvement; however, those with Modic type 2 changes did not improve further [61].

Truumees et al. conducted a prospective, open-label, single-arm study of 28 patients with CLBP and presenting with Modic type 1 or 2 changes who underwent radiofrequency ablation (RFA) of the BVN [62]. The patients demonstrated a significant improvement from baseline in ODI, VAS, SF-36, and EQ-5D-5L scores at 3 month post-operation (all *p* < 0.0001) [62]. Macadaeg et al. extended this work in a cohort of 45 patients with the same inclusion criteria followed up to 12 months [63]. All measured PROs (ODI, VAS, SF-36, EQ-5D-5L) showed statistically significant improvements from 6 weeks to 12 months [63]. These trials highlight substantial evidence for relief for patients with Modic type 1 or 2 receiving BVN ablation as measured by ODI, VAS, SF-36, and ED-5D-5L.

Khalil et al. conducted a multicenter, prospective randomized control study with a cohort of 140 patients (66 underwent BVN ablation and 74 served as controls) [46]. At the three-month follow-up, they found BVN was superior to the control group in function, pain, and quality of life as measured by ODI, VAS, SF-36 and EQ-5D-5L, respectively (all *p* < 0.001) [46]. In a follow-up study, Smuck et al. reported continued improvement 12 months post-operatively in all patient-reported measures for the BVN group (*p* < 0.001) [45]. Additionally, control patients who also underwent BVN ablation treatment demonstrated statistically significant improvement in all PRO measures at three and six months post-operatively (all *p* < 0.001) [45]. These studies reinforce the sustained benefit of BVN ablation in patients with CLBP with Modic type 1 and 2 changes.

Fischgrund et al. conducted a randomized sham-controlled trial on BVN ablation in patients with CLBP and Modic type 1 or type 2 changes [64]. At 3 months, the treatment group (147 patients) exhibited significant improvement in ODI scores (*p* = 0.019) compared to the control group (78 patients). However, VAS improvements in pain were only significant starting at the 6-month follow up (*p* = 0.008) [64]. In a follow-up study, ODI and VAS remained significantly improved, even at the 24-month mark (*p* < 0.001) [65]. The long-term analysis demonstrates the consistent benefit of BVN ablation in patients with CLBP and Modic type 1 and 2 changes in functionality and pain as measured by ODI and VAS, respectively. Markman et al. performed a post hoc analysis of the same trial population of the original Fischgrund et al. trial, finding that patients who decreased opioid use experienced significantly greater ODI and VAS improvements, but only in the BVN ablation treatment group (both *p* < 0.001) [66]. This analysis demonstrates that, in patients with Modic type 1 or 2 changes, BVN ablation may lead to pain relief and functional improvement, enabling opioid dosage reduction.

Schnapp et al. also performed BVN ablation in 16 patients with CLBP and Modic type 1 or type 2 changes as described by Fischgrund et al. [64,67]. Patients again showed significant improvements in function, pain, and quality of life as measured by ODI, VAS, and SF-36 scales at 1, 3, and 6-month follow-up (all *p* < 0.05) [67]. These trials all reinforce BVN ablation as a consistent and proven treatment for CLBP and function, but also for overall quality of life as captured by SF-36 and EQ-5D-5L.

## 6. Modic Change as Criteria for Pain Interventions

Historically, the intervertebral disc has been posited as the source of CLBP, contributing to radicular symptoms, which are typically characterized by pain that shoots down the leg, and disruption of sensation in a dermatomal distribution [68]. This concept of the intervertebral disc as a pain generator is reflected in the rates of elective spinal fusions and laminectomies, which have risen steadily in recent years [69]. Disc degeneration, however, is not always a pathologic process and occurs as a part of normal aging [70]. With an aging US population, it is critical to distinguish between age-related degenerative changes in the spine and true DDD or radiculopathy, which are pathologic and contribute to CLBP. The desire to better understand degenerative changes in the spine has led to studies that show that the origin of CLBP often has a vertebrogenic component and is rarely purely discogenic in nature [71]. Vertebrogenic pain has several clinical characteristics that distinguish it from other CLBP sources, including radiculopathy, SI joint dysfunction, and facet joint-associated pain. Vertebrogenic pain typically presents midline at the L3-S1 levels and has minimal radiation to the paraspinal or gluteal regions [72]. In contrast, SI joint pain typically radiates to the posterior superior iliac spines, and facet joint-associated pain typically presents in the paraspinal region rather than midline [73]. Vertebra-genic pain is typically worse with spinal flexion-based movements, such as bending forward, sitting for extended periods of time, or lifting objects from a bent-over position [43].

As a part of normal aging, vertebral endplates are subject to substantial stress during weight-bearing and activities of daily living [74]. Endplate damage is central to the underlying pathophysiology of vertebrogenic LBP, a clinical syndrome that correlates with the presence of MC on MRI [75,76]. Vertebral endplates are innervated by BVNs, which enter the vertebral body posteriorly via the basi-vertebral foramen [6]. Studies have demonstrated significant co-localization of calcitonin gene-related peptide (a neuropeptide involved in pain transmission) with protein gene product 9.5 (a neuronal marker) in histological sections, supporting a nociceptive role for nerves following the basi-vertebral distribution pattern [77].

As understanding of the nociceptive role of BVNs has progressed, therapeutic interventions have been developed to target the BVN and its role in vertebra-genic LBP [78]. Although the clinical presentation of vertebrogenic LBP is well characterized in the literature, symptoms alone are insufficient criteria for interventions and, thus, MCs have emerged as key imaging criteria for determining eligibility for interventions. Patients are candidates for BVN ablation if they have vertebrogenic LBP for a period greater than six months refractory to conservative treatments and evidence of Modic type 1 and/or type 2 changes on imaging at any level from L3-S1 [46,64,79,80,81]. Painful MCs typically affect the L4-L5 or L5-S1 levels, with Kuisma et al. finding that patients with CLBP are 2.28 times more likely to have MC at the L5-S1 vertebral level compared to those without CLBP [79,82]. A pooled analysis of three clinical trials conducted by Boody et al. demonstrated that CLBP of greater than 5 years duration and higher ODI scores at baseline were associated with greater odds of treatment success. Conversely, smoking history and lower ODI scores at baseline were associated with lower odds of treatment success [83]. These findings contribute to optimization of patient selection, as clinical and demographic factors can be considered alongside biomarkers such as MC to identify patients most likely to respond to BVN ablation.

Based on clinical characteristics and associated imaging findings guiding patient selection, BVN ablation employs a minimally invasive technique to target the BVN. During the procedure, a small incision is made on the patient’s back and a cannula is inserted through a needle into the vertebral body under fluoroscopic guidance. Once positioned near the BVN, RFA is performed to disrupt the nerve’s ability to transmit pain signals [84,85]. The goal is to provide long-term pain relief by targeting the nerve tissue within the vertebral body, without the need for more invasive surgical procedures [86].

Studies have shown promising results regarding safety and efficacy of BVN ablation for the treatment of vertebrogenic LBP. Becker et al. conducted the earliest known BVN ablation clinical study in 2017, in which they evaluated 17 patients undergoing BVN ablation and assessed outcomes via ODI to evaluate functional disability along with the VAS to evaluate patients’ pain levels [87]. Follow-up at 3 months demonstrated statistically significant decreases in ODI and VAS, with none of the patients experiencing serious adverse events related to the treatment [87]. The SMART trial conducted by Fischgrund et al. in 2018 is the largest BVN ablation study to date [64]. In a randomized, double-blinded, multicenter trial enrolling 225 patients, BVN ablation outperformed sham treatment as demonstrated by improvement in ODI scores at 3-month follow-up, with 76.5% of patients treated with BVN ablation reporting meaningful clinical improvement [64]. The INTRACEPT trial, a parallel, open-label, randomized control trial that enrolled 140 patients across 20 sites, demonstrated that BVN ablation led to significantly greater improvements in VAS and ODI at 3-month follow-up, prompting early crossover due to significant treatment efficacy [46]. While these studies demonstrate significant pain reduction and functional improvement for patients undergoing BVN ablation, further studies with long-term follow-up periods will continue to clarify the durability of clinical benefit and identify patients most likely to benefit from the procedure. Additionally, patients with osteoporosis, scoliosis, severe spinal stenosis, and spondylolisthesis > 2 mm were excluded from the original clinical trials [64]. Future research could explore the potential benefits of BVN ablation in patient populations typically excluded from initial studies, such as those with systemic or spinal infections, obesity, depression, radicular pain, disc protrusions greater than 5 mm, or a history of prior spinal surgery [46,64,79,80,81]. Currently, it is thought that the natural course of MC follows a progressively worse degeneration [88,89]. Huang et al. found a correlation between MC and local biomechanical stress that contributes to the distribution of MC and conversion of type 1 to type 2 in patients treated conservatively [88]. In addition, Jensen noted MC located from levels L4-S1 are more likely to extend further into the vertebra and along the endplate [89].

In 2024, Fogel et al. documented eight vertebral compression fractures (VCFs) among 74 patients who underwent BVN ablation over the course of one year. All eight patients who suffered VCFs had underlying osteoporosis, and the authors of this study acknowledged that the observed VCFs may have been incidental findings related to the patients’ underlying osteoporosis [81]. Therefore, an opportunity remains for investigation of the safety and efficacy of BVN ablation in patients with reduced bone density and complex spinal pathologies, with an emphasis on investigating the risk of VCFs related to BVN ablation.

Additionally, the biomechanical impact of BVN ablation on vertebral body integrity remains an important consideration. In 2020, De Vivo et al. conducted a study in which they performed a 3-month follow-up CT scan in 56 patients after BVN ablation to evaluate post-procedure bone mineral density [90]. A 0.5-cm^2^; region of interest was placed on pre-procedure CT scans within the central core of the vertebral body (the ablation target) and compared to 3-month post-procedure CT scans. Prior to BVN ablation, the mean bone density value of the region of interest was 95.6 Hounsfield Units (HUs). At 3-months post-procedure, the mean bone density value in the region of interest was 150.9 HU, representing a 57% increase. These findings suggest that BVN ablation may induce localized sclerosis or bone remodeling in the ablated region, potentially altering the biomechanical properties of the vertebral body [90]. However, the clinical implications of these changes remain uncertain, and future investigation may offer clarity on whether localized increases in bone mineral density confer a protective effect against VCF, or if such changes instead create new structural vulnerabilities in the vertebral body. As patient selection criteria for BVN ablation undergoes continued refinement, characterizing these structural changes and understanding the procedure’s impact on bone mineral density will be important for optimizing clinical outcomes and patient safety.

Alongside safety and clinical efficacy, cost-effectiveness is an important factor when considering broad adoption of BVN ablation as a treatment for CLBP. A model-based analysis using data from the INTRACEPT trial demonstrated an incremental cost-effectiveness ratio of USD 11,376 per quality-adjusted life year at a five-year horizon from introduction of the procedure. Modeling demonstrated a greater than 99% probability that BVNA is cost-effective in the US, based on commonly accepted willingness-to-pay thresholds of USD 100,000 to USD 150,000 per quality-adjusted life year. These emerging data suggest that BVN ablation is a cost-effective CLBP treatment compared to standard of care alone, and future studies can continue to elucidate the procedure’s long-term economic impact [91].

Overall, these studies highlight the clinical benefit and cost-effectiveness of BVN ablation in treating vertebrogenic LBP, with MCs serving as critical imaging biomarkers for identifying those most likely to benefit from the procedure, optimizing patient selection and improving clinical outcomes.

## 7. Conclusions

Modic type 1 and 2 changes have been utilized as meaningful visible biomarkers in the diagnosis and treatment of CLBP with vertebrogenic features. The association between MC and patient-reported outcomes is consistently demonstrated across multiple studies. Type 1 MC characterized by edematous and fibrovascular changes in vertebral endplate and Type 2 MC characterized by the replacement of hematopoietic marrow with fatty tissue demonstrate strong correlation with elevated pain intensity (VAS), functional impairment (ODI, RMQ), and worsening quality of life (SF-36 and EQ-5D-5L). Prospective trials show that patients with MC type 1 or 2 have sustained improvements in pain, function, and quality of life following BVN ablation. The replicable pattern of BVN ablation as a therapeutic treatment for patients with type 1 or 2 MC highlights the role of MC as a diagnostic and predictive biomarker for patients suffering from CLBP. Integrating MC assessments into the refinement of treatment plans may improve outcomes for patients with CLBP.

## Data Availability

No new data were created or analyzed in this study. Data sharing is not applicable to this article.

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
