# Peer review of "Modic Changes as Biomarkers for Treatment of Chronic Low Back Pain"

_biomedicines, 2025, doi:10.3390/biomedicines13071697_

Round 1
Reviewer 1 Report
Comments and Suggestions for Authors
This is a narrative review to propose Modic changes, especially type 1 and 2, as objective biomarkers of vertebrogenic chronic low back pain, suggesting their utility to serve as key diagnostic findings and outcome variables for BVN(basivertebral nerve) ablation procedure.
It is a very well written manuscript, covering almost all essential issues regarding Modic changes such as histologic, molecular and even genetic aspects, in a clear and neat way. The logical buildup throughout the manuscript is quite strong, starting from MR findings and pathology to strong relationships with clinical manifestation especially before and after BVN ablation.
I think this manuscript can be accepted in the current form, but I would suggest one major and two trivial issues to address as below:
One major issue is that I am concerned this article may give the readership a wrong impression that BVN ablation is a perfectly established procedure for vertebrogenic chronic LBP, which is not the case. There are not enough long term follow-up results reported yet. Considering most patients with chronic LBP live more than several decades years after the onset, following up for 3 months or two years may not be long enough to judge its safety and efficacy as is described in the section “6. Modic Change as Criteria for Pain Interventions”. For example, BVN ablation may cause neuropathic arthropathy - when the disc including endplates is regarded as a joint. Also, it is not clear whether it would outperform the natural course in the long run.
So, I would recommend the authors to add one or two sentences to implicate the idea something like — “BVN ablation appears safe and effective in short term follow-ups but it is not clear whether it would outperform the natural course in the long run. Being not a completely established treatment yet, it requires further basic researches and clinical trials of longer term follow-up”. I know this is not the main issue of this manuscript, but I believe that adding it up may strengthen the authors’ proposition further.
Two trivial issues:
1. In line 124, it reads “compared with a 0.3 T scanner”. Is “0.3T” a typo for “3.0T”?
2. In line 33, the authors defined “MC” as the abbreviation of “Modic changes” at the beginning of Introduction, but term “Modic changes” is still frequently used in the text along with the term “MC”. Consistency is needed.
Author Response
Thank you so much for taking the time and consideration to provide thorough and thoughtful feedback for this manuscript. Please find the detailed responses below and the corresponding revisions highlighted in the re-submitted manuscript.
Comment 1:
This is a narrative review to propose Modic changes, especially type 1 and 2, as objective biomarkers of vertebrogenic chronic low back pain, suggesting their utility to serve as key diagnostic findings and outcome variables for BVN(basivertebral nerve) ablation procedure.
It is a very well written manuscript, covering almost all essential issues regarding Modic changes such as histologic, molecular and even genetic aspects, in a clear and neat way. The logical buildup throughout the manuscript is quite strong, starting from MR findings and pathology to strong relationships with clinical manifestation especially before and after BVN ablation.
I think this manuscript can be accepted in the current form, but I would suggest one major and two trivial issues to address as below:
One major issue is that I am concerned this article may give the readership a wrong impression that BVN ablation is a perfectly established procedure for vertebrogenic chronic LBP, which is not the case. There are not enough long term follow-up results reported yet. Considering most patients with chronic LBP live more than several decades years after the onset, following up for 3 months or two years may not be long enough to judge its safety and efficacy as is described in the section “6. Modic Change as Criteria for Pain Interventions”. For example, BVN ablation may cause neuropathic arthropathy - when the disc including endplates is regarded as a joint. Also, it is not clear whether it would outperform the natural course in the long run.
So, I would recommend the authors to add one or two sentences to implicate the idea something like — “BVN ablation appears safe and effective in short term follow-ups but it is not clear whether it would outperform the natural course in the long run. Being not a completely established treatment yet, it requires further basic researches and clinical trials of longer term follow-up”. I know this is not the main issue of this manuscript, but I believe that adding it up may strengthen the authors’ proposition further.
Response 1: Thank you for pointing out how the manuscript could give the reader the wrong impression about BVN ablation. We agree with this comment, therefore we have added Sentences demonstrating the need for BVN ablation studies with longer follow-up and investigation of the procedure in a broader patient population than was studied in the original clinical trials. We appreciate the reviewer’s thoughtful feedback as these additions help contextualize existing evidence and highlight gaps in the literature that represent areas for future investigation. These changes can be found highlighted on page 8 paragraph 2, lines 352 to 364.
Comment 2: In line 124, it reads “compared with a 0.3 T scanner”. Is “0.3T” a typo for “3.0T”?
Response 2: We appreciate the careful review of our manuscript. The reference to “0.3 T scanner” is not a typographical error, as the meta-analysis from Herlin et al. includes studies that compare low-field MRI (0.3 Tesla) with standard 1.5 T scanners. These studies demonstrate that more Modic changes overall and more Modic type 2 changes but fewer type 1 changes were diagnosed in a 1.5 T scanner compared with a 0.3 T scanner, which is consistent with the statement in our manuscript.
Comment 3: In line 33, the authors defined “MC” as the abbreviation of “Modic changes” at the beginning of Introduction, but term “Modic changes” is still frequently used in the text along with the term “MC”. Consistency is needed.
Response 3: We appreciate the reviewer’s observation regarding the consistency of terminology. The manuscript has been revised to use the abbreviation “MC” after its initial definition. In addition, the same consistency of terminology was also revised for all the abbreviations listed at the bottom of page 9. For example, all "chronic low back pain" was updated to CLBP after the initial definition.
Reviewer 2 Report
Comments and Suggestions for Authors
- The manuscript notes that Modic changes types 1 and 2 exhibit significant correlations with symptom severity and predict favorable responses to basivertebral nerve ablation. It is crucial to incorporate a thorough discussion on the limitations of the existing evidence, including the variability in response rates, the long-term efficacy of the intervention, and the generalizability of the findings to diverse patient populations. Additionally, the authors should address the potential risks and complications linked to basivertebral nerve ablation.
- The article should offer a more comprehensive analysis of the cost-effectiveness and comparative effectiveness of basivertebral nerve ablation relative to other treatments for chronic low back pain. This information is essential for clinicians and policymakers when contemplating the adoption of this intervention in clinical practice.
Author Response
Thank you so much for taking the time and consideration to provide thorough and thoughtful feedback for this manuscript. Please find the detailed responses below and the corresponding revisions highlighted in the re-submitted manuscript.
Comment 1: The manuscript notes that Modic changes types 1 and 2 exhibit significant correlations with symptom severity and predict favorable responses to basivertebral nerve ablation. It is crucial to incorporate a thorough discussion on the limitations of the existing evidence, including the variability in response rates, the long-term efficacy of the intervention, and the generalizability of the findings to diverse patient populations. Additionally, the authors should address the potential risks and complications linked to basivertebral nerve ablation.
Response 1: We appreciate the thoughtful review of our manuscript, and we agree that discussion of the limitations of current evidence for BVN ablation will strengthen the paper. We have added sentences demonstrating the need for BVN ablation studies with longer follow-up and outlined the need to study this procedure in patients with reduced bone density and complex spinal pathologies. Additionally, we added a discussion of the potential risk for vertebral compression fractures related to BVN ablation and identified this as a key area for further study. These changes can be found highlighted on page 8 in paragraph 2 from lines 352 to 364.
Comment 2: The article should offer a more comprehensive analysis of the cost-effectiveness and comparative effectiveness of basivertebral nerve ablation relative to other treatments for chronic low back pain. This information is essential for clinicians and policymakers when contemplating the adoption of this intervention in clinical practice.
Response 2: We appreciate the thoughtful suggestion to include discussion of cost-effectiveness and we agree that discussion of the economic impact of BVN will strengthen our manuscript. In response, we have added a discussion of emerging data, including model-based analyses from the INTRACEPT trial demonstrating favorable cost-effectiveness profiles for BVN ablation. While direct comparative cost-effectiveness data remain limited in the current literature, we have acknowledged this gap and emphasized the need for future studies to further inform clinical and policy decision-making. This addition aims to provide clinicians and policymakers with a clearer understanding of the economic considerations relevant to the adoption of BVN ablation. These changes can be found highlighted on page 8, paragraph 3 from lines 365 to 374.
Response to comments on the Quality of English Language:
Response 1: We appreciate the feedback that the english could be improved more clearly to express the research. We agree with this comment therefore we have made revision throughout the manuscript with regards to grammar. In addition, we have also for consistency of terminology used the abbreviations outlined on the bottom on page 9 after its initial definition.
Round 2
Reviewer 2 Report
Comments and Suggestions for Authors
- The authors have acknowledged the need for longer follow-up studies to assess the long-term efficacy of basivertebral nerve ablation. It would be beneficial to include a discussion on the current state of knowledge regarding the natural history of Modic changes and how this might influence the long-term outcomes of treatment interventions.
- The manuscript could benefit from a more detailed analysis of the patient selection criteria for basivertebral nerve ablation. It is important to understand which patient subgroups are most likely to benefit from this intervention and whether there are specific characteristics that predict a positive response.
- The authors have mentioned the need for further study on the potential risk of vertebral compression fractures related to basivertebral nerve ablation. It would be valuable to discuss the current understanding of the biomechanical changes that occur post-ablation and how these might impact the integrity of the vertebral body.
- The manuscript could include a more comprehensive review of the literature on the pathophysiology of Modic changes, particularly focusing on the underlying mechanisms that link these changes to chronic low back pain. This could provide insights into potential new therapeutic targets.
- Given the manuscript's focus on Modic changes as biomarkers, it would be helpful to discuss the role of other imaging biomarkers in the context of chronic low back pain and how they compare with Modic changes in terms of diagnostic and prognostic utility.
Author Response
Thank you for taking the time to provide thoughtful feedback on our manuscript.Please find the detailed responses below and the corresponding revisions highlighted in the re-submitted manuscript.
Comment 1: The authors have acknowledged the need for longer follow-up studies to assess the long-term efficacy of basivertebral nerve ablation. It would be beneficial to include a discussion on the current state of knowledge regarding the natural history of Modic changes and how this might influence the long-term outcomes of treatment interventions.
Response 1: We appreciate the review of our manuscript, and we agree that expanding the discussion on the current literature regarding the natural history of Modic change and its impact on long-term outcomes will enhance the clarity of the manuscript. We have added sentences explaining the natural history of Modic change. These changes can be found highlighted on page 9 paragraph 2 from lines 404-409.
Comment 2:The manuscript could benefit from a more detailed analysis of the patient selection criteria for basivertebral nerve ablation. It is important to understand which patient subgroups are most likely to benefit from this intervention and whether there are specific characteristics that predict a positive response.
Response 2: We appreciate the thoughtful review of our manuscript. We agree that providing a more detailed description of the patient selection criteria would help differentiate who would benefit from basivertebral nerve ablation. We added sentences elaborating on the initial studies selection criteria and also mentioned patient subgroups who were not included. These changes can be found highlighted on page 8 paragraph 3 and 4 from lines 365-376 and also on page 9 paragraph 2 on lines 401-404.
Comment 3: The authors have mentioned the need for further study on the potential risk of vertebral compression fractures related to basivertebral nerve ablation. It would be valuable to discuss the current understanding of the biomechanical changes that occur post-ablation and how these might impact the integrity of the vertebral body.
Response 3: We appreciate the review of our manuscript. We agree that additional discussion on the biomechanical changes that occur post-ablation would enhance the clarity of the manuscript. We added a paragraph that describes the current understanding of the effects of basivertebral ablation on the vertebral body. These changes can be found highlighted on pages 9 paragraph 4 and page 10 paragraph 1, from lines 417 to 433.
Comment 4: The manuscript could include a more comprehensive review of the literature on the pathophysiology of Modic changes, particularly focusing on the underlying mechanisms that link these changes to chronic low back pain. This could provide insights into potential new therapeutic targets.
Response 4: We appreciate the thorough review of our manuscript. We also agree that a more detailed review of the pathophysiology of Modic change would elucidate the connection to chronic low back pain. We added an additional paragraph to thoroughly describe the pathophysiology. These changes can be found on page 5 paragraph 2 from lines 193 to 214.
Comment 5: Given the manuscript's focus on Modic changes as biomarkers, it would be helpful to discuss the role of other imaging biomarkers in the context of chronic low back pain and how they compare with Modic changes in terms of diagnostic and prognostic utility.
Response 5: We appreciate the review of our manuscript. We agree it would be beneficial to include a discussion on other imaging biomarkers used in chronic low back pain. We included various other imaging tools used in low back pain and a discussion of the predictive value of these other imaging biomarkers. These changes can be found highlighted on page 3 paragraph 9 and page 4 paragraph 1 from lines 136 to 145.
Round 3
Reviewer 2 Report
Comments and Suggestions for Authors
Thank you for your revision.